# Targeted Bioluminescent Imaging of Pancreatic Ductal Adenocarcinoma Using Nanocarrier-Complexed EGFR-Binding Affibody–Gaussia Luciferase Fusion Protein

**DOI:** 10.3390/pharmaceutics15071976

**Published:** 2023-07-19

**Authors:** Jessica Hersh, Yu-Ping Yang, Evan Roberts, Daniel Bilbao, Wensi Tao, Alan Pollack, Sylvia Daunert, Sapna K. Deo

**Affiliations:** 1Department of Biochemistry & Molecular Biology, Leonard M. Miller School of Medicine, University of Miami, Miami, FL 33136, USA; jhersh@miami.edu (J.H.); yyang22@med.miami.edu (Y.-P.Y.); sdaunert@med.miami.edu (S.D.); 2The Dr. John T. McDonald Foundation Bionanotechnology Institute, University of Miami, Miami, FL 33136, USA; 3Sylvester Comprehensive Cancer Center, Leonard M. Miller School of Medicine, University of Miami, Miami, FL 33136, USA; err92@med.miami.edu (E.R.); danielbilbao@med.miami.edu (D.B.); wtao@med.miami.edu (W.T.); apollack@med.miami.edu (A.P.); 4Department of Pathology and Laboratory Medicine, Leonard M. Miller School of Medicine, University of Miami, Miami, FL 33136, USA; 5Department of Radiation Oncology, Leonard M. Miller School of Medicine, University of Miami, Miami, FL 33136, USA

**Keywords:** targeted imaging, bioluminescent imaging, bioluminescent protein, fusion protein, EGFR targeting

## Abstract

In vivo imaging has enabled impressive advances in biological research, both preclinical and clinical, and researchers have an arsenal of imaging methods available. Bioluminescence imaging is an advantageous method for in vivo studies that allows for the simple acquisition of images with low background signals. Researchers have increasingly been looking for ways to improve bioluminescent imaging for in vivo applications, which we sought to achieve by developing a bioluminescent probe that could specifically target cells of interest. We chose pancreatic ductal adenocarcinoma (PDAC) as the disease model because it is the most common type of pancreatic cancer and has an extremely low survival rate. We targeted the epidermal growth factor receptor (EGFR), which is frequently overexpressed in pancreatic cancer cells, using an EGFR-specific affibody to selectively identify PDAC cells and delivered a *Gaussia* luciferase (GLuc) bioluminescent protein for imaging by engineering a fusion protein with both the affibody and the bioluminescent protein. This fusion protein was then complexed with a G5-PAMAM dendrimer nanocarrier. The dendrimer was used to improve the protein stability in vivo and increase signal strength. Our targeted bioluminescent complex had an enhanced uptake into PDAC cells in vitro and localized to PDAC tumors in vivo in pancreatic cancer xenograft mice. The bioluminescent complexes could delineate the tumor shape, identify multiple masses, and locate metastases. Through this work, an EGFR-targeted bioluminescent–dendrimer complex enabled the straightforward identification and imaging of pancreatic cancer cells in vivo in preclinical models. This argues for the targeted nanocarrier-mediated delivery of bioluminescent proteins as a way to improve in vivo bioluminescent imaging.

## 1. Introduction

In vivo imaging is an indispensable tool used in biomedical research and clinical practice for applications such as diagnostics, disease monitoring, the tracking of therapeutics, and understanding biological processes [1,2,3]. Current common imaging techniques include magnetic resonance imaging (MRI), computed tomography (CT), positron emission tomography (PET), ultrasound, and optical imaging, including fluorescence and bioluminescence [4]. The optimal method to use varies depending on factors such as the clinical application, biomedical research question, and resources. For investigating specific biological processes, molecular imaging with engineered reporter fusions that act as specific molecular probes, such as bioluminescent fusions, can provide a comprehensive solution, allowing for both tracking and defining the spatial location of specific cells and molecules.

Bioluminescent imaging (BLI) is ideal for in vitro and in vivo imaging because of its high sensitivity, high signal-to-noise ratio, nontoxicity, noninvasiveness, high-throughput capability, and low cost [5,6]. In some applications, the molecular imaging of specific targets provides more information than conventional methods, and BLI can be executed using specific molecular probes. BLI has been shown to be faster and more cost-effective than methods such as MRI and PET, because multiple mice can be imaged at once with shorter acquisition times. In addition, bioluminescent reporters are advantageous because they have a very low background signal compared to fluorescent reporters and do not require excitation [7,8].

Despite the high potential of BLI, there has been slow progress in its application for human studies due to challenges with emission in the blue region of the spectrum and reduced penetration in deep tissues. There have been many innovative approaches reported for overcoming these problems. For example, the bioluminescent/fluorescent resonance energy transfer (BRET/FRET) to redshift the signal enables greater tissue penetration, even if the efficiency of the energy transfer is low and has reduced sensitivity [9,10,11]. Mutational studies of bioluminescent proteins emitting in the far-red region of the spectrum have also been conducted with some success [12,13,14]. The majority of BLI studies, thus far, have been based on transfecting cells with plasmids expressing bioluminescent proteins to achieve localized bioluminescent signals; however, this is limited to in vitro and small rodent studies, and can require days to achieve a localized signal that may also induce immunogenic effects [1,15]. Thus, there is still an opportunity to expand molecular optical imaging using bioluminescence principles and proteins.

We hypothesized that the creation of a bioluminescent molecular fusion that allows for the specific recognition of disease markers and enables the targeted delivery of a bioluminescent reporter, combined with a nanocarrier-based delivery for higher loading of molecular fusion, would provide an enhanced BLI (Figure 1). To the best of our knowledge, a similar methodology has only been explored by Han et al. with an ErbB2-targeted bioluminescent protein on a liposomal nanocarrier used for monitoring metastatic ovarian cancer [16].

To test our hypothesis, we used pancreatic ductal adenocarcinoma (PDAC) as a disease model. PDAC is the most common type of pancreatic cancer, accounting for more than 90% of cases [17]. PDAC has an extremely high mortality [18] rate, with the estimated 5-year relative survival rate in the United States being 12%–the lowest of all cancer types [19]. The high mortality is partially attributed to the frequent late diagnosis of the disease, which has been reported to occur in 90% of cases [20]. Studies have shown that EGFR is overexpressed in as many as 89–95% of pancreatic cancer cases [21,22], making it a good candidate for PDAC detection. Targeted imaging can improve upon current diagnostic methods by specifically identifying and locating pancreatic cancer cells.

In this work, we engineered an EGFR-targeting bioluminescent fusion protein that could be loaded onto a G5-polyamidoamine (G5-PAMAM) dendrimer nanocarrier for PDAC detection. To enable specific targeting and bioluminescent signal generation, we used an EGFR-binding affibody (ZEGFR) as a PDAC-targeting ligand and *Gaussia* luciferase (GLuc) as a bioluminescent reporter. Affibodies are antibody-mimetic proteins that have the specificity of antibodies but that are a much smaller size, facilitating their synthesis and improving their stability [23,24]. We showed that the resulting fusion protein–dendrimer complex localized PDAC cells in vitro and in vivo and enabled BLI. This opens the door for potential applications of BLI by creating fusions with other recognition moieties that bind disease biomarkers of interest.

## 2. Materials and Methods

### 2.1. Plasmid Construction

The pCold I plasmid was purchased from Clontech (Clontech Laboratories, Inc., Mountain View, CA, USA), and the DNA anti-EGFR-tagged *Gaussia* luciferases were synthesized from GenScript (GenScript Biotech Corporation, Pscataway, NJ, USA). The DNA was cloned into plasmid pCold I and then transformed into *E. coli* NEB 5-alpha cells (New England Biolabs, Ipswich, MA, USA). The sequences of constructed plasmids were confirmed (Genewiz, South Plainfield, NJ, USA) and the plasmids were transformed into *E. coli* SHuffle (New England Biolabs) for protein expression.

### 2.2. Expression and Purification of Recombinant Proteins

The bacterial cells containing plasmids were grown overnight at 37 °C, 250 rpm in 5 mL of LB broth containing 100 μg/mL ampicillin. This culture was inoculated into 500 mL of TB broth containing 100 μg/mL ampicillin and incubated at 37 °C, 250 rpm until the OD600 reached 1.0. The culture medium was cooled in an ice-water bath for over 1 h and, subsequently, isopropyl β-D-1-thiogalactopyranoside (IPTG) was added to the culture medium at the final concentration of 0.1 mM. After incubating 24–48 h at 16 °C, the cells were harvested through centrifugation at 7000× *g* for 10 min and were lysed in 10 mL of BugBuster reagent (EMD Millipore, Burlington, MA, USA). The fusion proteins were purified using a column of Ni-NTA agarose (Qiagen, Hilden, Germany), according to the manufacturer’s instructions. The recombinant proteins were subject to 4–20% SDS-PAGE and confirmed with Coomassie brilliant blue staining. The proteins were finally dialyzed against 1X phosphate-buffered saline (PBS, pH 7.4) and the concentrations were determined using a protein assay kit (Bio-Rad Laboratories, Hercules, CA, USA).

### 2.3. G5-PAMAM–ZEGFR-GLuc Complex Formulation

To form the complexes, 50 µg of purified protein (0.002 µmol ZEGFR-GLuc fusion protein or 0.0027 µmol GLuc alone) was mixed in 1X PBS (pH 7.4) at a 3:1 molar ratio with a generation 5 polyamidoamine (G5-PAMAM, Twentyfirst Century Biochemicals, Inc., Marlboro, MA, USA) dendrimer (0.0007 µmol G5-PAMAM mixed with ZEGFR-GLuc or 0.0009 µmol G5-PAMAM mixed with GLuc). The components were combined at room temperature, vortexed for 5 s and incubated at room temperature for 30 min. The complexes were created fresh for each experiment and always used within 1 h. The bioluminescent light intensity was measured with a CLARIOstar Plus Microplate Reader (BMG Labtech, Ortenberg Germany) for all experiments, as described in Section 3.1. The size was measured with a Zetasizer NanoZS (Malvern Panalytical, Malvern, UK).

### 2.4. Cell Lines and Cell Lysate Preparation

The PANC 10.05 cell line (ATCC, Manassas, VA, USA) was grown with ATCC-formulated RPMI-1640 Medium (Gibco, Thermo Fisher Scientific, Waltham, MA, USA) supplemented with 10 units/mL human recombinant insulin (Sigma-Aldrich, St. Louis, MO, USA), 15% fetal bovine serum (Thermo Fisher Scientific, Waltham, MA, USA) and 1X Pen Strep Glutamine (Gibco, Thermo Fisher Scientific, Waltham, MA, USA). A human pancreatic duct epithelial cell line (H6c7, Kerafast, Boston, MA, USA) was grown with keratinocyte serum-free media (Gibco, Thermo Fisher Scientific, Waltham, MA, USA) supplemented with the included human recombinant epidermal growth factor and bovine pituitary extract, as well as 1X Pen Strep Glutamine (Gibco, Thermo Fisher Scientific, Waltham, MA, USA). The HEK 293T cell line (ATCC, Manassas, VA, USA) was grown with Dulbecco’s Modified Eagle Medium (Gibco, Thermo Fisher Scientific, Waltham, MA, USA) supplemented with 10% fetal bovine serum and 1X Pen Strep Glutamine (Gibco, Thermo Fisher Scientific, Waltham, MA, USA). All cells were maintained in humidified incubators at 37 °C with 5% CO_2_.

To prepare the cell lysates for Western blotting (Appendix A), the cells were grown in a 10 cm plate and allowed to grow to 80% confluence. At this time, the cells were lysed with a Peirce RIPA buffer (Thermo Fisher Scientific, Waltham, MA, USA) containing a 1X Halt protease inhibitor (Thermo Fisher Scientific, Waltham, MA, USA). The plate was washed twice with cold 1X PBS (pH 7.4) followed by the addition of 100 µL of the RIPA buffer with a protease inhibitor. The cells were scraped off the plate and transferred into a microcentrifuge tube. Here, they incubated while on ice for 30 min. After this time, the tube was spun at 13,000 RPM for 30 min at 4 °C, and the supernatant containing the proteins was transferred to a clean microcentrifuge tube and stored at −80 °C until use.

### 2.5. EGFR Selectivity in Cells

To check the EGFR selectivity of the fusion protein, 250 µL of PANC 10.05 cells (EGFR-positive) and H6c7 cells (EGFR-negative) was added to microcentrifuge tubes at a concentration of 1000 cells/mL in their respective complete media. The fusion protein was added to each tube at a concentration of 400 nM and incubated at room temperature. Following incubation, the tubes were spun down at 150× *g* for 5 min and the supernatant was removed. The cells were washed three times with 1X DPBS (Gibco, Thermo Fisher Scientific, Waltham, MA, USA) and resuspended in 1X DPBS following the washes. The suspensions were transferred to a 96-well plate and luminescence was measured with a CLARIOstar Plus Microplate Reader (BMG Labtech, Ortenberg, Germany). The luminometer was programmed to inject a coelenterazine substrate (NanoLight Technology, Prolume Ltd., Pinetop-Lakeside, AZ, USA) to a final concentration of 0.46 mg/mL and measure the intensity (wavelength 480 +/− 20) for 3 s.

### 2.6. Fluorescent Tagging and Microscopy

The ZEGFR-GLuc proteins, GLuc proteins and G5-PAMAM dendrimers were fluorescently tagged using Alexa Fluor-NHS ester dyes (AF594, AF488, Thermo Fisher Scientific, Waltham, MA, USA). The G5-PAMAM dendrimers were tagged with AF488 and all proteins were tagged with AF594. The dyes were mixed with the proteins or dendrimer at a 12:1 molar ratio in a 3 mL reaction vessel on a stir plate, covered from light, for 2 h at room temperature. For each reaction, 250 µL of protein/dendrimer was used and the volume of dye was adjusted to achieve the 12:1 dye:protein/dendrimer ratio. Following the 2 h reaction, 200 µL of a quenching buffer (25 mM Tris-HCl, 150 mM NaCl, pH 7.2, Tris-HCl from Roche, Basel Switzerland, NaCl from VWR International Radnor, Wayne, PA, USA) was added to the reaction vessel and stirred for an additional 1 h at room temperature while being covered from light. After this, the reaction solution was removed and dialyzed using a 3500 MWCO Slide-A-Lyzer dialysis cassette G2 (Thermo Fisher Scientific, Waltham, MA, USA) against 1X PBS (pH 7.4). The solution was protected from light during dialysis. The solution was dialyzed for three days and the 1X PBS (pH 7.4) was refreshed three times. At the end of dialysis, the reaction solution was carefully removed from the cassette and transferred to a clean microcentrifuge tube, where it was stored at 4 °C and protected from light.

For the fluorescent microscopy experiments, 8-chamber Culture Slides (Falcon Corning, Corning, NY, USA) were seeded with 50,000 PANC 10.05 cells/well in 500 µL media. Throughout this experiment, the slides and treatments were kept in as dark a setting as possible. Cells were grown for 48 h in a humidified incubator at 37 °C with 5% CO_2_. After this time, fresh complexes were created using the above methods (Section 2.3), but with the fluorescently tagged proteins and dendrimers. Each chamber treatment contained 3.25 µg protein and complexes were formed at the molar ratio described above. The complexes were diluted in media to a final volume of 500 µL. All media were aspirated from the 8-chamber slide and each treatment was added to its respective well. The treatments were incubated for 3 h, after which all media were again removed from the well and 500 µL of buffered 10% formalin (pH 6.8–7.2, VWR International Radnor Pennsylvania USA) was added for fixation. The cells were fixed for 10 min at room temperature, after which the wells were washed 3 times with 1X PBS (pH 7.4) to remove excess formaldehyde and stop the fixing reaction. Next, the chambers were incubated with 0.5 µg Hoechst 33342 trihydrochloride, trihydrate dye (Thermo Fisher Scientific, Waltham, MA, USA) in 500 µL of 1X PBS (pH 7.4) for nuclear staining. The dye was incubated for 5 min at room temperature, followed by 3 washes with 1X PBS (pH 7.4) to remove excess dye, followed by one wash with water to remove salt from the PBS. Then, all liquid was removed from the chambers and the frame was removed. The slide was dried and a Fluoromount Aqueous Mounting Medium (Sigma-Aldrich, St. Louis, MO, USA) was added to the slide prior to adhering a coverslip. This was allowed to dry for approximately 15 min, after which the edges of the coverslip were sealed with regular transparent nail polish. The sealed slide was allowed to dry fully prior to imaging. The slides were imaged with a fluorescent microscope (Keyence, Osaka, Japan) using the brightfield function and fluorescence microscopy with laser wavelengths of 350 nm, 488 nm and 595 nm. ImageJ (National Institutes of Health, public domain license) was used for the quantification of the fluorescence intensity.

### 2.7. Mouse Models and Animal Studies

All animal protocols were approved by the University of Miami Institutional Animal Care and Use Committee (protocols 21-073 and 21-028). Female NSG (NOD Scid gamma) mice were used to develop the xenografts using PANC 10.05 cells. There were 5 mice per treatment group for the experiments described in Section 3.3 and Section 3.4, with 1 mouse from each treatment group used for the ex vivo analysis described in Section 3.4. For the treatments, the complexes were prepared as described above (Section 2.3) and each treatment contained 50 µg of protein. The mice were shaved prior to imaging. The mice received treatments via tail-vein injection, and at either 3 h or 6 h after receiving the protein treatments, the mice received 100 µg of water-soluble coelenterazine (NanoLight Technology, Prolume Ltd., Pinetop-Lakeside, AZ, USA) substrate via tail-vein injection. All injections had a fixed volume of 100 µL. Immediately following the substrate injection, the mice were imaged using the IVIS^®^ Spectrum in vivo imaging iystem (PerkinElmer, Waltham, MA, USA). Isoflurane was used to anesthetize the mice for the imaging. For the experiment in Section 3.5, a nude mouse (Foxn1nu; *n* = 1) was used and received intratumoral injections with fixed volumes of 100 µL. All image analyses were performed and signal intensities measured using the Living Image software (PerkinElmer, Waltham, MA, USA).

### 2.8. Statistics

GraphPad Prism (GraphPad, La Jolla, CA, USA) was used for the statistical analysis. The data in Section 3.1 were normalized by setting the maximum value as 100 and leaving 0 as 0. The normalized data were unitless. The in vitro experiments were performed in triplicate. A *p*-value of *p* ≤ 0.05 was considered significant. For the two-group comparisons, a two-tailed, unpaired Student’s *t*-test was used. For the multiple comparisons, one-way ANOVA was used. For the post hoc analysis, Tukey tests were performed when *n* was the same for each group, and Fisher’s LSD test was used when *n* was different. To compare two curves, the global fitting tool of GraphPad Prism was used.

For the animal studies, a power calculation was performed with G*Power (Universität Düsseldorf, Düsseldorf, Germany) to determine the number of mice needed per treatment group based on in vitro data (α = 0.05; β = 0.05; effect size d = 2.9). The analysis determined that each group would need *n* = 4 mice with an actual power of 0.975.

## 3. Results and Discussion

### 3.1. Development of a Fusion Protein and Protein–Dendrimer Complex

Our work aimed to utilize cell-specific targeting and nanocarrier delivery to increase the amount of protein reaching the imaging location and improve the signaling output for bioluminescence. Utilizing targeting molecules with nanocarrier-based delivery to directly deliver bioluminescent proteins to cells or tissues of interest would both increase the signal in that area and drastically reduce the time needed to achieve a localized signal compared to cell transfection. We first selected *Gaussia* luciferase (GLuc) because it is an attractive bioluminescent protein, as it is one of the smallest (18.2 kDa) and brightest of the luciferase family, having a reported 1000-fold or greater signal intensity compared to other luciferases such as firefly or *Renilla* luciferase [25,26,27]. Our group previously demonstrated that GLuc is advantageous over other luciferases because of its temperature stability and high signal output [8,28]. We further enhanced these characteristics by introducing mutations that prolonged the bioluminescence and enhanced the light stability [26,29].

We then identified EGFR as a suitable target for specifically identifying PDAC cells, since EGFR has been shown to be overexpressed in 89–95% of PDAC cases [21,22] and is associated with carcinogenesis [30]. Targeted therapy for EGFR is one of few treatment options that has shown a significant survival benefit for PDAC [31,32]. We identified an opportunity to utilize EGFRs for localizing imaging agents to pancreatic cancer cells. For EGFR-targeting, an EGFR-specific affibody (ZEGFR) [33] was utilized.

To develop the fusion protein, a plasmid was designed containing sequences for an EGFR-binding affibody (ZEGFR) and the GLuc bioluminescent protein (Figure 2A, sequence available in Appendix A). Following the expression of the protein in bacterial cells, the proteins were extracted and purified. The purified proteins were used to confirm the EGFR-binding specificity by incubating EGFR-negative human pancreatic ductal epithelial (H6c7) cells and EGFR-positive PANC 10.05 cells with the fusion proteins and measuring the bioluminescent intensity of each. The bioluminescent signal was significantly greater in the EGFR-positive cells, indicating the binding specificity of the proteins to EGFRs (Figure 2B). The bioluminescent emission spectrum (Figure 2D) of the ZEGFR-GLuc fusion protein was slightly redshifted from that of native GLuc. This redshift may have been caused by slight changes in the secondary structure of GLuc upon the fusing to the EGFR-binding affibody. Additionally, the bioluminescent kinetic curve (Figure 2E) showed that the signal decay was much faster in the native GLuc compared to the ZEGFR-GLuc, as expected [26,29]. The bioluminescent kinetic curve of the mutant GLuc showed a slight increase before a gradual decrease, demonstrating the prolonged signal compared to native GLuc (Figure 2E).

Nanocarriers can be used to form supramolecular structures that increase the binding of cargo and allow for tailored delivery, such as pH dependence [34]. Nanocarriers are increasingly being used to improve imaging resolution and quality in other methods, such as MRI and fluorescence imaging [35], and, thus, can similarly improve bioluminescent image quality. We selected the G5-PAMAM dendrimer, an ultrasmall, monodisperse nanoparticle with a diameter of 5.4 nm [36]. The PAMAM dendrimer is known to protect cargo from degradation, increase the amount of protein reaching the PDAC cells by increasing circulation time, and aid in localization with the enhanced permeability and retention (EPR) effect. PAMAM dendrimers are known to bind proteins through electrostatic interactions, forming supramolecular complexes [37]. The pI for the fusion protein was 5.725, and, therefore, was negatively charged at pH 7.4, allowing it to bind with the positively charged dendrimers. Additionally, our group previously demonstrated that enhanced biocompatibility can be achieved through surface modification with organic molecules such as proteins and peptides [38,39].

The obtained fusion protein could be complexed with a PAMAM dendrimer nanocarrier to form a targeted complex by taking advantage of the abundant surface amine groups of the dendrimer, enabling the functionalization of the nanocarrier with the fusion protein. This complex could then localize ZEGFR bioluminescent fusion proteins to pancreatic cancer cells in vitro and in vivo for the identification and imaging of the cells (Figure 1). The size of the complex, as determined with the use of dynamic light scattering, was 306.7 ± 31.6 nm. The complex formation of the dendrimer and the ZEGFR-GLuc fusion protein did not affect the emission spectra of the GLuc (Figure 2C). However, the complex formation significantly affected the signal decay by slowing the decay and prolonging the signal strength (Figure 2F, *p* < 0.0001, Appendix A).

The EGFR selectivity, demonstrated with the PANC 10.05 cells (Figure 2B), supports the efficacy of this platform and methodology. The efficacy of this strategy is also supported by similar studies, which utilized EGFR targeting to enhance the delivery of both imaging agents and therapeutic molecules in PDAC [40,41,42,43,44,45,46]. Our unique methodology differed from the imaging studies in numerous ways. For instance, the imaging modalities used in the similar studies included fluorescence imaging [40] and MRI [46]. While these methods are effective, they have some limitations and may not be ideal for every research application, encouraging the development of alternative solutions such as the one presented here. Although not used as frequently as CT, MRI produces the best images and may allow for earlier detection than CT; however, it is slightly limited due to the very high cost and sometimes requires contrast agents [47,48,49]. Fluorescence imaging is used extensively in preclinical models, and indocyanine green has been approved by the Federal Drug Agency since 1956 [50]. As with any method, there are some limitations, such as the need for lasers for excitation, which can cause photobleaching and toxicity, and potential autofluorescence [51,52,53]. By utilizing bioluminescence imaging, our methodology provides researchers with more options for their work.

Another way in which our methodology differed was through the use of an EGFR-targeting affibody instead of an anti-EGFR antibody such as cetuximab [40] or a single-chain anti-EGFR antibody [46]. Affibodies are much smaller than antibodies and single-chain antibodies. Cetuximab is a large protein approximately 150 kDa in size [54]. The single-chain anti-EGFR antibody is smaller with a size of 25 kDa [46]; however, the EGFR-targeting affibody is smallest at 8.1 kDa [55]. While all these targeting moieties have a high affinity for EGFR, there are some applications where a smaller size may be beneficial. For example, a study in glioma imaging found that fluorescently labeled EGFR-targeted affibodies had a significantly better delineation of tumor margins compared to fluorescently labeled cetuximab [54]. The smaller size may increase efficacy in some applications, making this tool a useful option for researchers.

### 3.2. In Vitro Delivery in Pancreatic Cancer Cells

PANC 10.05 is an EGFR-positive pancreatic adenocarcinoma cell line (Appendix A) that was used to demonstrate the localization enhancement of the complex using fluorescent microscopy. The GLuc bioluminescent protein was incubated with PANC 10.05 cells either alone, complexed to a dendrimer, fused to the EGFR-targeting affibody, or fused to the targeted affibody and complexed to a dendrimer. The localization to the cells was significantly enhanced when fused to the targeting affibody and complexed to the dendrimer (Figure 3B). The GLuc protein alone and GLuc complexed to the dendrimer (without EGFR-targeting) had minimal localization to the cells and adhered to the slide surface in small clusters (Figure 3A, rows two and three). In contrast, when fused to the targeting protein, localization was only seen in the cells (Figure 3A, row four). The greatest localization, however, was seen when the fusion protein was complexed to a dendrimer, resulting in a strong signal in the cells and clearly showing the margins of the cells (Figure 3A, row five).

When comparing the average ratio between the signal from the fluorescently tagged proteins and from the stained nuclei for each treatment condition, the targeted complex had significantly more protein localizing to the cells than the other treatments (Figure 3B). The nuclei signal was used to normalize the protein signal to the number of cells in the well. The targeted protein alone also localized the cells significantly more when compared with the nontargeted protein, but less than the targeted dendrimer complex. These studies supported the use of the fusion protein–dendrimer complex for the in vivo imaging of PDAC cells, and clearly demonstrated the improvement EGFR targeting had upon the localization of the cells of interest.

The utilization of biological ligands, such as proteins, peptides, antibodies and affibodies, to target specific cells is a commonly used technique. We previously reviewed extensively the benefits and applications of using peptide ligands to direct polymers, such as PAMAM dendrimers, to specific cells [56]. Additional reviews have noted similar conclusions and described enhancement in selective delivery when nanoparticles are modified with targeting ligands. The targeting ligands are typically biomolecules [57,58,59]. Similarly, PAMAM dendrimers were noted to be good candidates for the delivery of many molecules, including proteins [60,61,62,63]. They have abundant surface functional groups, are monodispersed, increase bioavailability and are known to protect cargo from degradation in pharmaceutical and biomedical delivery applications [39,58,62,64,65]. As such, many groups have used them to deliver and protect cargo such as genes, peptides, and proteins, finding that their nanoparticle formulations enhance the stability of the cargo. For example, in one study, a group found that using a PAMAM-based nanoparticle to deliver the brain-derived neurotrophic factor (BDNF) protein helped overcome half-life challenges with the protein [66]. In another, a group overcame a similar challenge with the protein angiotensin 1–7 using hydroxyl-terminated PAMAM dendrimers [67].

Protein–dendrimer complexes allow the targeting and functionalization of the dendrimers using proteins and prevent protein degradation using dendrimers. This work took advantage of these benefits by utilizing a biological ligand to target PDAC cells and a PAMAM dendrimer to protect the targeting ligand and the bioluminescent cargo by means of fusing the proteins together and complexing them with the dendrimer.

### 3.3. In Vivo Delivery in Pancreatic Cancer Xenograft Mouse Models

Xenograft mouse models were developed using PANC 10.05 cells to generate NSG mice with pancreatic cancer tumors that were EGFR-positive (Appendix A). Mice were injected intravenously with protein (GLuc, EGFR-targeted GLuc (fusion protein), or EGFR-targeted GLuc complexed with dendrimers; *n* = 5 per treatment group), and 3 h or 6 h later, the bioluminescent substrate coelenterazine was injected intravenously prior to imaging with IVIS. Three different types of xenograft models were generated—subcutaneous, renal capsule, and orthotopic—and the targeted fusion protein complexed with the dendrimer was able to localize to the tumors in all three models (Appendix A). The complex only localized when tumors were present; in the absence of a tumor, low levels of signal were detected nonspecifically in the mouse (Appendix A).

In vivo studies confirmed our previous in vitro findings, indicating that the complex formation with the dendrimer facilitated tumor localization. When comparing the signal intensity among the three treatment conditions (GLuc alone, fusion protein, and fusion protein complexed to a dendrimer) at 3 h and 6 h post-protein injection, the fusion protein complexed to the dendrimer had a higher bioluminescent intensity than the fusion protein and nontargeted protein (Figure 4A). In the representative images shown in Figure 4A, the fusion protein complexed to the dendrimer had a smaller change in signal intensity compared to the other treatments (Appendix A). The targeted complex signal decreased by 25%, while signal from the fusion protein alone decreased by 36%.

To facilitate a comparison, during each imaging session, mice with different treatments were imaged at the same time (as shown in Figure 4A). The quantified signal intensity of the control, targeted protein, and targeted complex were then determined and quantified (Figure 4B). Several factors could affect the bioluminescent light intensity that were not related to the amount of bioluminescent protein. For example, Virostko et al. demonstrated that simply changing the angle between the camera and the animal can alter the signal intensity [68]. Due to this, a direct comparison of the bioluminescent signal between each imaging session was not possible. Instead, for each imaging session, the signal intensity at the tumor site for the three different treatments was normalized to that of the targeted complex, and the targeted complex signal was set as 100%. There were five mice per treatment group (for a total of five separate imaging sessions). The signal intensities analyzed were taken 3 h after the treatment with the bioluminescent proteins and 8 min after the injection of the coelenterazine substrate to control for incubation times. The normalized signals were then averaged and compared. The results showed that the targeted fusion protein was essential for achieving a high signal, as the nontargeted dendrimer-complexed GLuc consistently showed a low signal. Complexing the targeted fusion protein to the nanocarrier further increased the signal intensity (Figure 4B).

Altogether, the results from these in vivo studies showed that the targeted fusion protein greatly enhanced localization in vivo and that complex formation with the dendrimer could further enhance this. The complex formation with the dendrimer alone was not enough to improve the signal because the native GLuc complexed to a nanocarrier did not have the same high signal (Figure 4B). As the nontargeted control in this study was conjugated to a dendrimer, we could be confident that the enhanced localization was due to the active targeting of the ZEGFR protein and not passive targeting from the dendrimer and EPR effect. This showed that the EGFR-targeting affibody was required to identify the PDAC cells and the dendrimer would enhance the signal output by concentrating the fusion protein at the targeted location.

### 3.4. In Vivo Tumor Margin Delineation and Location of Metastatic Cancer Cells

After demonstrating the efficacy of the platform in vivo, the ability of the targeted protein and complex to identify tumor margins and locate metastatic cells was evaluated. When the ultrasound-confirmed tumors were small (~100 mm^3^), the renal-capsule-xenografted mice were treated with the fusion protein, dendrimer-complexed fusion protein, or nontargeted GLuc protein complexed with a dendrimer. The targeted complex was able to identify two separate masses at the tumor site in one mouse, which correlated with the two separate masses seen with ultrasound (Appendix A, left panel, and Appendix A). This indicated that the targeted complexes could delineate the tumor margin and identify individual clusters of cells, making it useful in imaging studies for detecting multiple masses and visualizing the shape and spread of tumors. As the tumors grew, the experiment was repeated to monitor the change in shape. The shape changed accordingly with changes in the tumor, enabling the monitoring of tumor growth (Appendix A). This could be useful for tracking treatment efficacy or monitoring tumor progression because the targeted proteins and targeted complexes localized in tumors of all sizes.

When the tumors were large and following the imaging of the tumors in vivo, the mice were sacrificed for analysis of the organs and the biodistribution of the bioluminescent proteins (Figure 5I–K; *n* = 1 per treatment group). The following organs were examined for each mouse: intestines, brain, heart, lungs, liver, spleen, left kidney, femur, and right kidney (which was used for the renal capsule xenograft). The bioluminescent intensity was measured for each organ and the average radiance was normalized to the signal at the tumor site (Figure 5A). It is important to note that by the end of this study, the tumor in the nontargeted control mouse had grown to nearly double and triple the size of the tumors of the other mice (tumor volumes are noted in the bottom right corner of Figure 5C–E). The large size of the tumor and the associated increased vasculature would result in an enhancement of the passive targeting due to the EPR effect, increasing the signal found in the tumor. Therefore, although the nontargeted control had a high signal at the tumor site, this may be explained by the large size of the tumor. By analyzing the radiance of the tumor normalized to the volume of the tumor, the signal was lower in the nontargeted control compared to the targeted treatments (Figure 5B). It is also important to note that the nontargeted mouse displayed high signals in other organs, including the intestines and heart (Figure 5E). This demonstrated that without EGFR-targeting, an off-site signal would occur.

An interesting result was the significant signal detected in the lungs of the mouse that received the targeted complex treatment. This signal was nearly equal in strength to that of the tumor from that mouse, and there was no other relevant detectable signal seen in the other organs (Figure 5A,D). The mouse that received the targeted protein treatment also had a slightly increased signal in the lungs (Figure 5A,C). This suggested metastatic pancreatic cancer cells in the lungs, a common site for pancreatic cancer metastases. To confirm it, the lungs and tumors of each mouse were sliced and stained to check for pancreatic cancer cells. Extensive metastases were found in the lungs, as indicated with H&E staining (Figure 5F–H, left images), revealing that the lungs of all the mice were invaded by metastatic PDAC cells. Although the mouse that received the nontargeted complex treatment had similar levels of metastases in the lungs, the bioluminescent signal was comparable to that of the heart and intestines and not of the tumor, indicating lower detection of the metastases (Figure 5A,E). This biodistribution study showed that the targeted complexes could be used to identify pancreatic cancer metastases, a useful feature for a cancer that is frequently diagnosed in its late stages. The ability to locate cancer cells throughout the body would aid in determining tumor grade and may help guide treatment decisions.

Taken together, the results of this study demonstrated the feasibility of using targeted complexes to delineate tumor margins in small and large tumors, and to identify metastases in vivo. The implications of these studies were that the targeted complexes may be useful in diagnosis, localization and tumor tracking in vivo.

### 3.5. Additional Capabilities and Applications of Targeted Complexes

In one final test, the potential of the targeted complex to produce a bioluminescent signal that can be detected with a standard camera was assessed. To achieve this, one nude mouse with a subcutaneous tumor was treated with 100 µg of the fusion protein complexed to dendrimer via an intratumoral injection, and 3 h later received 100 µg of the coelenterazine substrate. Immediately following the injection, the room was darkened and an image was captured with an average smartphone. As can be seen in Appendix A, the signal intensity was strong enough that an image could be captured without the use of any special equipment. This further supported the choice to use *Gaussia* luciferase as a strong bioluminescent reporter. This also suggested that the methodology may be useful for quickly confirming a biopsy or checking if any cancerous cells remain after a surgical resection in a conventional surgical room setup. This application of fluorophores is already being explored as a tool in surgical settings [69] and bioluminescent reporters may serve in this function in the near future.

## 4. Conclusions

In this study, a platform was developed that utilized EGFR targeting to bring the bioluminescent protein *Gaussia* luciferase to pancreatic ductal adenocarcinoma cells for imaging and identification by using a dendrimer as a nanocarrier. The targeting affibody was fused to the bioluminescent protein and the resulting fusion protein was complexed to a polyamidoamine dendrimer to improve the stability and signal strength of the molecule. The targeted complex localized PDAC cells in vitro and in vivo, and was capable of identifying the tumor margin, tracking tumor growth, and locating metastases in xenograft mouse models. The signal strength was strong enough to be detected without specialized equipment and captured with a smartphone. The targeted complex could have translational applications in the diagnosis and monitoring of tumor progression. The methodology utilized could be expanded to further applications, making it an adaptable tool for bioluminescence-based imaging. For example, this platform, without any modification, could be useful for other cancers that overexpress EGFR. Alternatively, a similar fusion protein could be developed with a different targeting ligand and could be easily adapted to dendrimers for new applications. With specifically targeting designer peptides constantly being developed [70,71], there are countless possibilities for adapting this system. In this work, we demonstrated a method that enabled in vivo bioluminescence-based imaging in mice, adding new possibilities to a rapidly growing field.

## Figures and Tables

**Figure 1 pharmaceutics-15-01976-f001:**
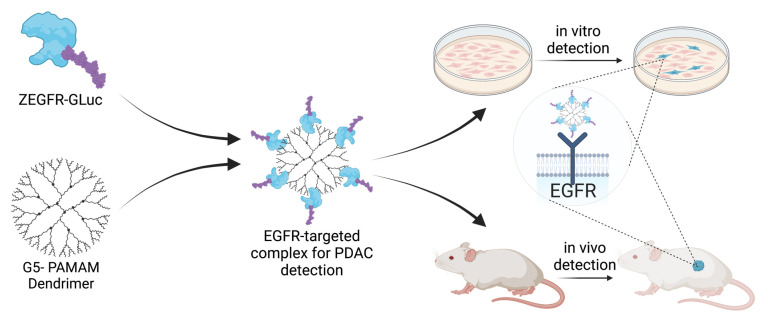
Formulation of the targeted complex and principle of the technology. The ZEGFR-GLuc protein was mixed with PAMAM dendrimer to form the targeted complex. The targeted complex selectively finds pancreatic cancer cells using the EGFR-targeting affibody both in vitro and in vivo, and delivers the bioluminescent protein *Gaussia* luciferase to those cells. This enables the selective identification and imaging of tumor cells.

**Figure 2 pharmaceutics-15-01976-f002:**
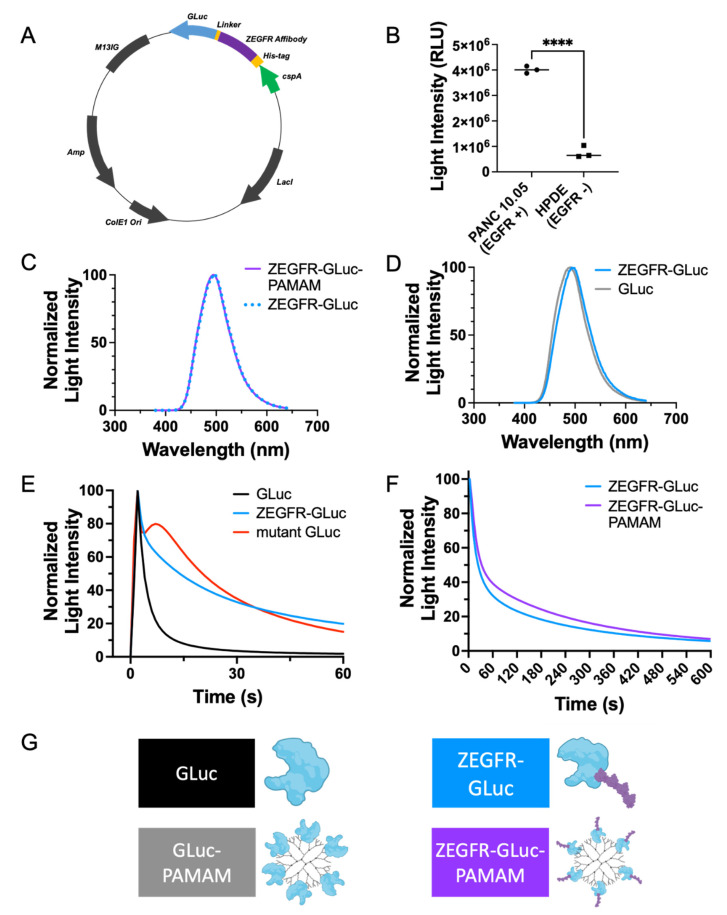
Characterization of the ZEGFR-GLuc–PAMAM complex. (**A**) Plasmid map for the ZEGFR-GLuc fusion protein. (**B**) The EGFR-targeted protein preferentially bound EGFR-positive PANC 10.05 cells compared to EGFR-negative HPDE cells. (**C**) Emission spectra of the fusion protein with and without dendrimer nanocarrier. The dendrimer did not shift the emission spectra. (**D**) Emission spectra of the fusion protein and native GLuc. The fusion protein had a slightly redshifted spectra. (**E**) Bioluminescent kinetics of the fusion protein, GLuc mutant and GLuc. The signal decay was much slower for the fusion protein compared to the native GLuc. (**F**) Bioluminescent kinetics of the fusion protein with and without dendrimer. The signal decay was slower when the fusion protein was bound to the dendrimer nanocarrier. (**G**) Color key with representation of the proteins for all graphs in this figure (**** *p* ≤ 0.0001; unpaired, 2-tailed Student’s *t*-test).

**Figure 3 pharmaceutics-15-01976-f003:**
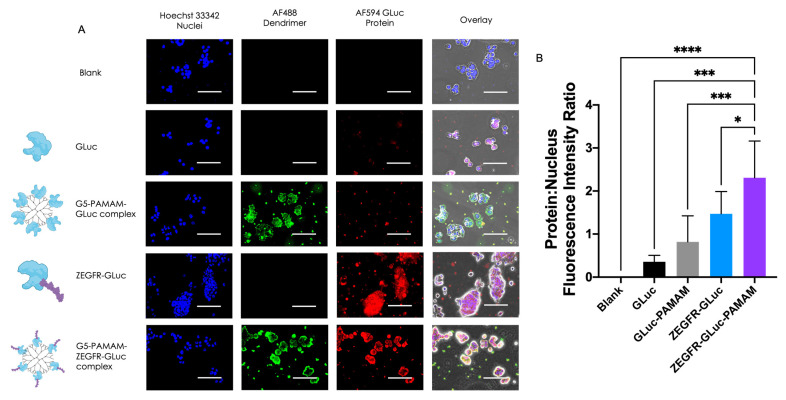
In vitro uptake of proteins in EGFR-positive PANC 10.05 cells. (**A**) Representative fluorescent microscopy images demonstrating localization enhancement with EGFR-targeting and binding to the dendrimer in EGFR-positive cells. In the second and third rows, no EGFR targeting was used and there was minimal binding to the cells. In the fourth row, the EGFR targeting improved localization to the cells compared to no targeting, but the addition of the dendrimer (fifth row) had the most significant enhancement in localization. Scale bar = 100 µm. Magnification 40×. (**B**) Quantification of the fluorescent signal from the tagged proteins. (* *p* ≤ 0.05; *** *p* ≤ 0.001; **** *p* ≤ 0.0001; one-way ANOVA with Fisher’s LSD test).

**Figure 4 pharmaceutics-15-01976-f004:**
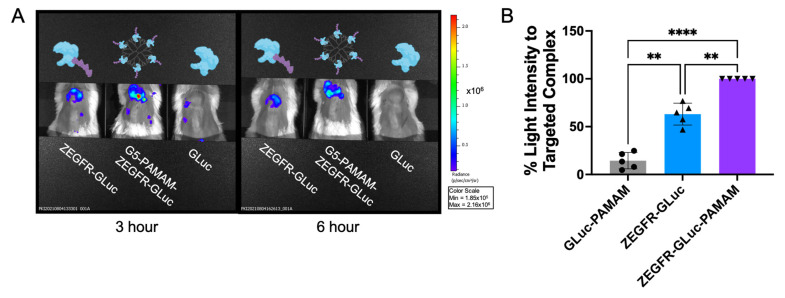
Complexing the fusion protein to a dendrimer resulted in a greater bioluminescent intensity in vivo at the site of the tumor. (**A**) Representative images of bioluminescent signal in pancreatic cancer xenograft mice at 3 and 6 h post-treatment. (**B**) In vivo bioluminescent signals normalized to the intensity of the ZEGFR-GLuc–PAMAM complex showing that the EGFR targeting greatly enhanced the signal at the tumor site. *n* = 5 mice per group (** *p* ≤ 0.01; **** *p* ≤ 0.0001; row-matched one-way ANOVA with Tukey test).

**Figure 5 pharmaceutics-15-01976-f005:**
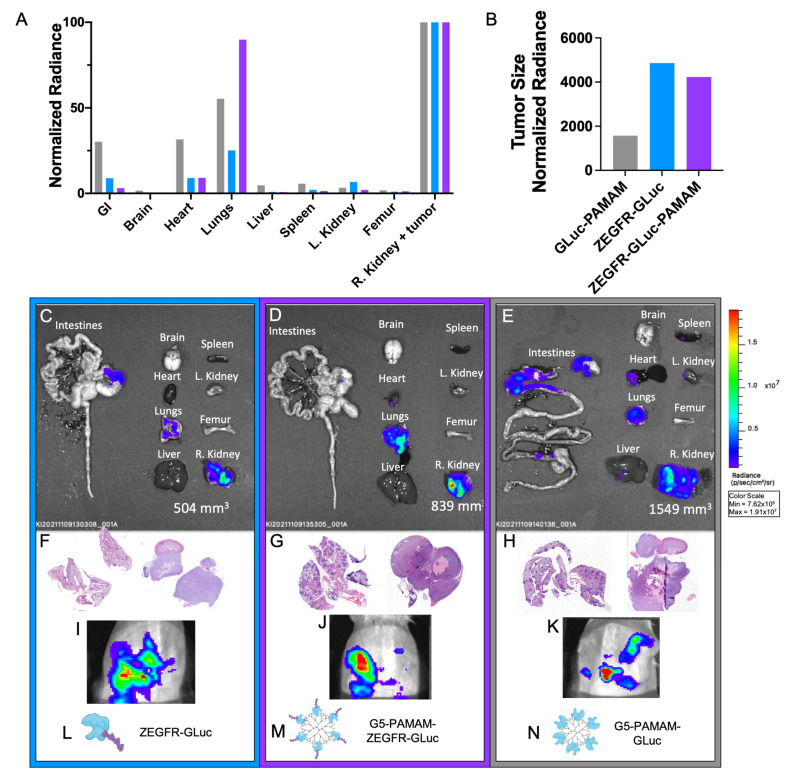
The targeted complexes were able to identify both localized and metastatic cancer cells in xenograft pancreatic cancer mouse models with renal capsule tumors (*n* = 1 per treatment group). (**A**) Quantified bioluminescent intensities in various organs for each mouse treatment. (**B**) Quantified bioluminescent intensities in the tumor normalized to the tumor volume. (**C**–**E**) Bioluminescent images of the various organs for each mouse treatment. Bioluminescent signal was only detected in the lungs and the right kidney, where the tumor grew, for the mouse that received the targeted complex, while the nontargeted complex had high signals in the right kidney, lungs, heart, and intestines. The targeted protein had slight off-target signals. (**F**–**H**) H&E-stained tissue slices of the lungs (left image) and right kidneys (right image) from the organs in (**B**) for the fusion protein, targeted complex, and control, respectively. Metastatic pancreatic cancer cells were identified through the darker purple tissues. (**I**–**K**) IVIS bioluminescent images of the tumor sites prior to sacrifice for the mice treated with the fusion protein, targeted complex, and control, respectively. (**L**–**N**) Visualization of the three treatments used, indicating which treatment was represented in each column.

## Data Availability

Data are contained within the article or Appendix A.

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
