# Peer review of "Targeted Bioluminescent Imaging of Pancreatic Ductal Adenocarcinoma Using Nanocarrier-Complexed EGFR-Binding Affibody–Gaussia Luciferase Fusion Protein"

_pharmaceutics, 2023, doi:10.3390/pharmaceutics15071976_

Round 1
Reviewer 1 Report
1. Abstract: It is well written and précised. Authors are advised to change the second sentence because it is a very general statement and casually written.
2. The introduction is perfectly written and very well structured.
3. Line 84-85: Statistical data must be obtained from recent surveys or references.
4. The materials and method section is also well organized.
5. Figure-1: EGFR-based active targeting is the main strategy here. Therefore, authors can point to the EGFR tagging in the targeted complex for PDAC detection.
6. The first two paragraphs of 3.1 discuss the strategic plan for BLI, and that can be shifted to the introduction, and authors can give one or two sentences before 3rd paragraph starts.
7. Figure 2C & D: Y-axis title should be changed to Normalized BL intensity (a.u.) or normalized emission intensity (a.u.), or Intensity (a.u.). Anyone can be used instead of light intensity. In the case of 2E and F also, the axis title can be taken care of if it is fluorescence-based because it is not mentioned in the result.
8. The authors spoke about the redshift and that they can explain it in terms of change in the microenvironment around the probe after binding.
9. The authors have stated that the vehicle was utilized to improve the stability (Line 34-35), but it was also said that the protein is highly stable in temperature and light without further modifications (Line 265-268). Please explain.
10. Did the authors check for the stability of the complex before and after the nanoformulation?
11. Line 535-536: EGFR overexpression is associated with several cancer types, including PDAC. The authors have also stated that the protocol, without any changes, can detect other cancer types too - then how the work was developed dedicated to a targeted diagnosis for PDAC.
12. What is the fate of the prepared complex? How does it gets eliminated from the bodily system?
13. Decision: Minor revision.
Well written and understandable.
Reviewer 2 Report
The authors developed a platform, which utilized epidermal growth factor receptor (EGFR) as a target to bring the bioluminescent protein Gaussia luciferase to pancreatic ductal adenocarcinoma (PDAC) cells for imaging and identification by using dendrimer as a nanocarrier. The targeted complex localized to PDAC cells in vitro and in vivo was reported to be capable of identifying the tumor margin, tracking tumor growth, and locating metastases in xenograft mouse models. The paper successfully contributed to the rapidly growing field of bioluminescence-based imaging in mice.
I have the following minor suggestions for improvement:
1. I did not found information on the polydispersity and toxicity of the dendrimers (G5-PAMAM) used for the construction of the nanocarriers. For commercial applications, this information will be very important.
2. What is the type of bond between the dendrimer and GLuc? Is it a covalent bond or another type? Similarly, it will be interesting for the reader to know explicitly the bond between ZEGFR and GLuc.
3. The authors may find helpful a recent paper on a similar anticancer platform based on the utilization of designed peptides: Mitochondrial Voltage-Dependent Anion Channel 1–Hexokinase-II Complex-Targeted Strategy for Melanoma Inhibition Using Designed Multiblock Peptide Amphiphiles, ACS Appl. Mater. Interfaces 2021, 13, 30, 35281–35293.
Reviewer 3 Report
The review of the manuscript „Targeted Bioluminescent Imaging of Pancreatic Ductal Adenocarcinoma Using Nanocarrier-complexed EGFR-binding Affibody-Gaussia Luciferase Fusion Protein” by Hersh et al.
The manuscript aims to discover a bioluminescent probe which could specifically target pancreatic ductal adenocarcinoma cancer cells. The manuscript is written well, however, I have several remarks which are as follow:
- what about the toxicity of dendrimers and their new bioluminescent probe? Are there any preliminary studies sone in this field?
- the selection of mice species should be included. How many animals were used in this study? More details are needed in in vivo paragraph.
Reviewer 4 Report
Jessica Hersh and colleagues present a quality and well-written experimental manuscript describing the targeted bioluminescent imaging of pancreatic ductal adenocarcinoma using nanocarrier-complexed EGFR-binding affibody-Gaussia luciferase fusion protein.
Authors have engineered an EGFR-targeting bioluminescent fusion protein which can be loaded onto a G5-polyamidoamine dendrimer nanocarrier for PDAC detection. To enable specific targeting and bioluminescent signal generation, they used an EGFR-binding affibody as a PDAC-targeting ligand and Gaussia luciferase as a bioluminescent reporter.
Authors showed that the resulting fusion protein-dendrimer complex localized to PDAC cells in vitro and in vivo and enabled BLI. This opens the door for potential applications of BLI by creating fusions with other recognition moieties that bind disease biomarkers of interest.
To achieve these results authors targeted epidermal growth factor receptor, which is frequently overexpressed in pancreatic cancer cells, using an EGFR-specific affibody to selectively identify PDAC cells, and delivered a Gaussia luciferase bioluminescent protein for imaging by engineering a fusion protein with both the affibody and the bioluminescent protein. This fusion protein was then complexed with a G5-PAMAM dendrimer nanocarrier. The dendrimer was used to improve protein stability and increase signal strength. Their targeted bioluminescent complex had enhanced uptake into PDAC cells in vitro and localized to PDAC tumors in vivo in pancreatic cancer xenograft mice. The bioluminescent complexes could delineate tumor shape, identify multiple masses, and locate metastases.
Authors showed that an EGFR-targeted bioluminescent-dendrimer complex enabled straightforward identification and imaging of pancreatic cancer cells in vivo in preclinical models. This argues for targeted nanocarrier-mediated delivery of bioluminescent proteins as a way to improve in vivo bioluminescent imaging.
Finally, authors conclude that they developed a platform which utilized EGFR-targeting to bring the bioluminescent protein Gaussia luciferase to pancreatic ductal adenocarcinoma cells for imaging and identification by using dendrimer as a nanocarrier. The complex has translational applications in diagnostics and monitoring of tumor progression.
Overall, the manuscript is highly valuable for the scientific community and should be accepted for publication.
===========
Other comments:
1) Please check for typos throughout the manuscript.
2) Please improve the images and tables where appropriate.
3) With regards to supramolecular nanocarrier complexes (dendrimers) - authors are kindly encouraged to cite the following article that describes a branched macrocyclic delivery system that has high relevance for fusion protein nanocarrier complexes. DOI: 10.1016/j.molliq.2022.120807
